# The effect of weight loss following 18 months of lifestyle intervention on brain age assessed with resting-state functional connectivity

Gidon Levakov[1]*[†], Alon Kaplan[2,3][†], Anat Yaskolka Meir[2], Ehud Rinott[2], Gal Tsaban[2], Hila Zelicha[2], Matthias Blüher[4], Uta Ceglarek[4], Michael Stumvoll[4], Ilan Shelef[5], Galia Avidan[6][‡], Iris Shai[2,4,7][‡]

[1]Department of Brain and Cognitive Sciences, Ben-Gurion University of the Negev, Beer Sheva, Israel; [2]The Health & Nutrition Innovative International Research Center, Faculty of Health Sciences, Ben-Gurion University of the Negev, Beer Sheva, Israel; [3]Department of Internal Medicine D, Chaim Sheba Medical Center, Ramat-Gan, Israel; [4]Department of Medicine, University of Leipzig, Leipzig, Germany; [5]Department of Diagnostic Imaging, Soroka Medical Center, Beer Sheva, Israel; [6]Department of Psychology, Ben-Gurion University of the Negev, Beer Sheva, Israel; [7]Department of Nutrition, Harvard T.H. Chan School of Public Health, Boston, United States

*For correspondence:
gidonle@post.bgu.ac.il

[†]These authors contributed equally to this work
[‡]These authors also contributed equally to this work

## Abstract

**Background:** Obesity negatively impacts multiple bodily systems, including the central nervous system. Retrospective studies that estimated chronological age from neuroimaging have found accelerated brain aging in obesity, but it is unclear how this estimation would be affected by weight loss following a lifestyle intervention.

**Methods:** In a sub-study of 102 participants of the Dietary Intervention Randomized Controlled Trial Polyphenols Unprocessed Study (DIRECT-PLUS) trial, we tested the effect of weight loss following 18 months of lifestyle intervention on predicted brain age based on magnetic resonance imaging (MRI)-assessed resting-state functional connectivity (RSFC). We further examined how dynamics in multiple health factors, including anthropometric measurements, blood biomarkers, and fat deposition, can account for changes in brain age.

**Results:** To establish our method, we first demonstrated that our model could successfully predict chronological age from RSFC in three cohorts (n=291;358;102). We then found that among the DIRECT-PLUS participants, 1% of body weight loss resulted in an 8.9 months' attenuation of brain age. Attenuation of brain age was significantly associated with improved liver biomarkers, decreased liver fat, and visceral and deep subcutaneous adipose tissues after 18 months of intervention. Finally, we showed that lower consumption of processed food, sweets and beverages were associated with attenuated brain age.

**Conclusions:** Successful weight loss following lifestyle intervention might have a beneficial effect on the trajectory of brain aging.

**Funding:** The German Research Foundation (DFG), German Research Foundation - project number 209933838 - SFB 1052; B11, Israel Ministry of Health grant 87472511 (to I Shai); Israel Ministry of Science and Technology grant 3-13604 (to I Shai); and the California Walnuts Commission 09933838 SFB 105 (to I Shai).

## Editor's evaluation

This study is indeed a landmark work that reports the significant benefits of lifestyle intervention in terms of attenuation of brain age and improvement in several tissue-based biomarkers. The findings from this study are of compelling and convincing nature that would encourage and support structured lifestyle intervention as an inclusive part of public health.

## Introduction

Brain aging is a complex, multifaceted process with various manifestations in different periods of the human lifespan, brain regions, and imaging modalities (*Jack et al., 2017*; *Bethlehem et al., 2022*). Nevertheless, reducing this complex process to a single scalar, the predicted brain age, may capture multiple conditions and risk factors associated with deviation from the normal aging trajectory (*Cole and Franke, 2017*). Brain age estimation is typically done by predicting chronological age from neuro-imaging data in a healthy training group of subjects and applying the fitted model to a new, unseen individual. This procedure enables estimating a measure of brain age independent of the individual's chronological age. Over-estimation of brain age, in relation to chronological age, is observed in several neurological conditions such as mild cognitive impairment, Alzheimer's disease (AD), schizophrenia, and depression (*Liem et al., 2017*; *Koutsouleris et al., 2014*; *Bashyam et al., 2020*), and is associated with an increase in mortality rate (*Cole et al., 2018*). Similarly, over-estimation of brain age was also found in obesity (*Franke et al., 2014*; *Kolenic et al., 2018*; *Ronan et al., 2016*), suggesting that the brain age framework may provide a powerful tool for assessing accelerated brain aging due to excessive weight. Critically, it is unclear whether dietary lifestyle interventions may have a beneficial, attenuative effect on the brain aging process.

Obesity is associated with multiple adverse health impacts also observed in normal aging (*Salvestrini et al., 2019*; *Tam et al., 2020*). These comorbidities of obesity and typical aging include the risk of cardiovascular disease (*Park et al., 2013*), inflammation (*Frasca et al., 2017*), type 2 diabetes (*Ahima, 2009*), DNA damage (*Niedernhofer et al., 2018*; *Shimizu et al., 2014*), and neurodegenerative processes (*Pugazhenthi et al., 2017*). The link between excessive weight and neuronal damage is likely mediated by adiposity, metabolic dysfunction, and alteration in the gut microbiome (*Gupta et al., 2020*; *Farruggia and Small, 2019*). These, in turn, promote inflammatory metabolic processes in the central nervous system (*Leigh and Morris, 2020*). Accordingly, reduction in gray and white matter volume (*Kullmann et al., 2015*; *García-García et al., 2019*), changes in brain connectivity (*Parsons et al., 2022*; *Daoust et al., 2021*), cognitive impairment (*Yang et al., 2018*), and the prevalence of dementia (*Pedditzi et al., 2016*) were all associated with midlife obesity. These anatomical (*Bethlehem et al., 2022*), functional (*Sala-Llonch et al., 2015*), and behavioral (*Fjell et al., 2017*) findings are also observed during normal aging. An increase in life expectancy (*Chang et al., 2019*) along with a sharp growth in obesity rates (*Abarca-Gómez et al., 2017*) elicit the need to characterize, treat, and perhaps prevent obesity-related brain aging.

We previously found that weight loss, glycemic control, lowering of blood pressure, and increment in polyphenols-rich food were associated with an attenuation in brain atrophy (*Kaplan et al., 2022*). Obesity is also manifested in aging-related changes in the brain's functional organization as assessed with resting-state functional connectivity (RSFC). These changes are dynamic (*Honey et al., 2007*) and can be observed in short time scales (*Bassett et al., 2011*) and thus of relevance when studying lifestyle intervention. Studies have linked obesity with decreased connectivity within the default mode network (*Doucet et al., 2018*; *Beyer et al., 2017*) and increased connectivity with the lateral orbitofrontal cortex (*Parsons et al., 2022*), which are also seen in normal aging (*Sala-Llonch et al., 2015*; *Lopez et al., 2020*). Longitudinal trials have reported changes in these connectivity patterns following weight reduction (*McFadden et al., 2013*; *Lowe et al., 2019*), indicating that they can be altered. However, findings regarding functional changes are less consistent than those related to anatomical changes due to the multiple measures (*Rubinov and Sporns, 2010*) and scales *Mišić and Sporns, 2016* used to quantify RSFC. Hence, focusing on a single measure, the functional brain age may better capture these complex changes and their relation to aging.

Here, as a sub-study of the Dietary Intervention Randomized Controlled Trial Polyphenols Unprocessed Study (DIRECT-PLUS *Yaskolka Meir et al., 2021b*), we examined the effect of successful weight loss following 18 months of lifestyle intervention on brain aging attenuation (*Figure 1*). We assessed

**eLife digest** Obesity is linked with the brain aging faster than would normally be expected. Researchers are able to capture this process by calculating a person's 'brain age' – how old their brain appears on detailed scans, regardless of chronological age. This approach also helps to monitor how certain factors, such as lifestyle, can influence brain aging over relatively short time scales. It is not clear whether lifestyle interventions that promote weight loss can help to slow obesity-driven brain aging.

To answer this question, Levakov et al. studied 102 individuals who met the criteria for obesity and took part in a lifestyle intervention aimed to improve diet and physical activity levels over 18 months. The participants received a brain scan at the beginning and the end of the program; additional tests and measurements were also conducted at these times to capture other biological processes impacted by obesity, such as liver health.

Levakov et al. used the brain scans taken at the start and end of the study to examine the impact of the lifestyle intervention on the aging trajectory. The results revealed that a reduction in body weight of 1% led to the participants' brain age being nearly 9 months younger than the expected brain age after 18 months. This attenuated aging was associated with changes in other biological measures, such as decreased liver fat and liver enzymes. Increases in liver fat and production of specific liver enzymes were previously shown to negatively impact brain health in Alzheimer's disease. Finally, examining more closely the food consumption reports completed by participants showed that reduced consumption of processed food, sweets and beverages were linked to attenuated brain aging.

The findings show that lifestyle interventions which promote weight loss can have a beneficial impact on the aging trajectory of the brain observed with obesity. The next steps will include determining whether slowing down obesity-driven brain aging results in better clinical outcomes for patients. In addition, the work by Levakov et al. demonstrates a potential strategy to evaluate the success of lifestyle changes on brain health. With global rates of obesity rising, identifying interventions that have a positive impact on brain health could have important clinical, educational and social impacts.

brain age based on RSFC taken before and after the intervention. Brain aging attenuation was quantified as the difference between the expected and observed brain age after the intervention. We trained and validated the age prediction model using two separate cohorts (n=291 [*Nooner et al., 2012*], 358 [*Shafto et al., 2014*; *Taylor et al., 2017*]), then applied it to our group of participants from the DIRECT-PLUS (n=102). We hypothesized that a successful reduction in anthropometric measurements following the intervention would attenuate brain aging. We then examined how multiple clinical outcomes, including liver, glycemic, lipids, and magnetic resonance imaging (MRI) fat deposition markers, would be related to attenuated brain aging. Finally, we report the correlation between brain age attenuation and changes in reported food consumption. To the best of our knowledge, this is one of the first studies that examined the beneficial effect of successful weight loss on the brain aging trajectory in humans, assessed by resting-state fMRI.

## Methods

In line with (*Simmons et al., 2012*) 21-word solution, we report how we determined our sample size, all data exclusions, all manipulations, and all measures in the study.

### Dataset used for training and validating the brain age model

Training, validation, and testing of the brain age model were conducted on data from two cohorts that included functional and structural brain MRI. The training was conducted on the enhanced Nathan Kline Institute (NKI)-Rockland Sample (*Nooner et al., 2012*) and testing on the Cam-CAN dataset (*Shafto et al., 2014*; *Taylor et al., 2017*). The NKI dataset is composed of 291 subjects (226 females, 65 males) recruited from Rockland County, USA. All participants provided informed consent and the study was approved by the Institutional Review Board at the Nathan Kline Institute

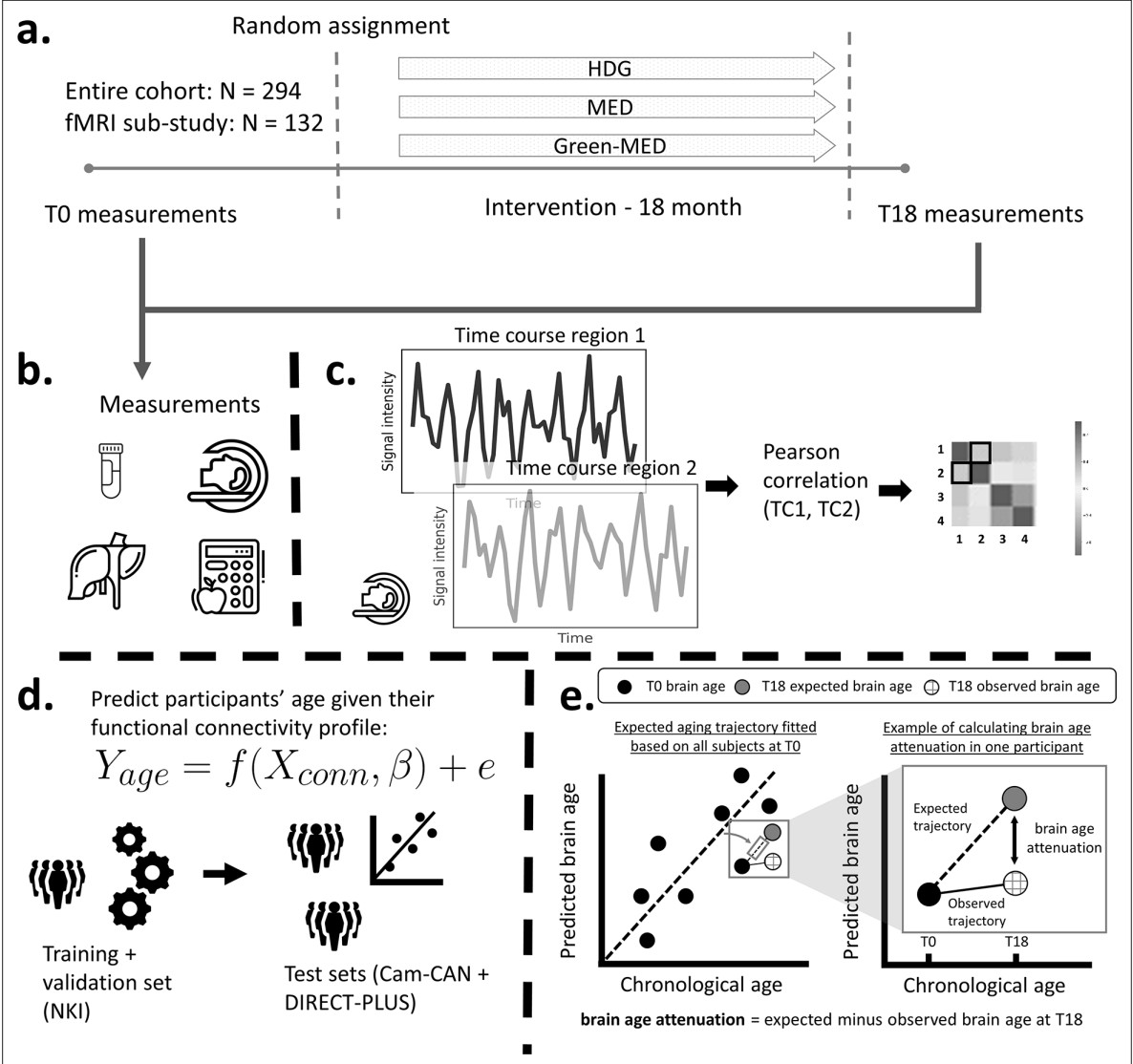

**Figure 1.** Study design and workflow. The Dietary Intervention Randomized Controlled Trial Polyphenols Unprocessed Study (DIRECT-PLUS) trial examined the effect of successful weight loss following 18-month lifestyle intervention on adiposity, cardiometabolic, and brain health across intervention groups. (**a**) Participants in the functional connectivity sub-study (N=132) completed the baseline measurements at T0. They were randomly assigned to three intervention groups: healthy dietary guidelines (HDG), an active control group, Mediterranean diet (MED), and green-MED. All groups were combined with physical activity (PA). Eighteen months following intervention onset, all measurements were retaken (T18). (**b**) Measurements included anthropometric measurements, blood biomarkers, fat deposition, and structural and functional brain imaging. (**c**) Functional brain imaging was conducted while subjects were at rest and was used to estimate resting-state functional connectivity (RSFC). RSFC measures the correlation between the time series of pairs of brain regions. (**d**) We fitted a linear support vector regression to predict chronological age from all pairwise correlations. We fitted the model on the Nathan Kline Institute (NKI) dataset, then tested and applied it to the Cambridge Centre for Ageing and Neuroscience (Cam-CAN) and the DIRECT-PLUS data. (**e**, left scatter plot) Based on the T0 data, we first computed the expected aging trajectory as the linear relation between the chronological and predicted age of all subjects. The fitted line represents the increase in the predicted age in relation to chronological age in the absence of an intervention. (**e**, right scatter plot) The fitted line was used to estimate the expected brain age at T18, given each participant's T0 brain age and the time passed between the T0 and T18 magnetic resonance imaging (MRI) scans. We computed the observed brain age by applying the brain age model to the T18 scans. Brain age attenuation was calculated as the expected brain age minus the observed at T18.

(#226781 and #239708) and Montclair State University (#000983A and #000983B). The Cam-CAN dataset includes 358 (193 females, 165 males) subjects roughly uniformly distributed from Cambridge City, UK. All participants provided informed consent, and the study was approved by the local ethics committee, Cambridgeshire 2 Research Ethics Committee (reference: 10/H0308/50). In both datasets,

we included only subjects within the DIRECT-PLUS age range (34–82 years). Exclusion criteria included unsuccessful completion of the preprocessing and quality control stages (see MRI preprocessing).

## Study design

This work was based on a sub-study of the DIREC-PLUS trial (clinicaltrials.gov ID: NCT03020186). The primary aims of the DIRECT-PLUS trial were 18-month changes in VAT, intrahepatic fat, and adiposity across intervention groups. The results for the primary outcomes were presented in separate publications (*Yaskolka Meir et al., 2021b*). The DIRECT-PLUS was launched in May 2017 and conducted in an isolated workplace in Israel (Nuclear Research Center Negev, Dimona, Israel). Most clinical and medical measurements, including anthropometric measurements, blood drawing, and lifestyle intervention sessions, were performed on-site. Among 378 volunteers, 294 met age (30+ years of age) and abdominal obesity inclusion criteria waist circumference (WC): men >102 cm, women >88 cm (*U.S. Department of Health and Human Services, 2013*; *Centers for Disease Control and Prevention, 2020*) or dyslipidemia (TG >150 mg/dL and high-density-lipoprotein-cholesterol [HDL-C] ≤40 mg/dL for men, ≤50 mg/dL for women [*Grundy et al., 2005*]). Exclusion criteria were inability to perform physical activity (PA); serum creatinine ≥2 mg/dL; serum alanine aminotransferase or aspartate aminotransferase more than three times above the upper limit of normal; a major illness that might require hospitalization; pregnancy or lactation; active cancer, or chemotherapy treatment in the last 3 years; warfarin treatment; pacemaker or platinum implantation; and participation in a different trial. Among 294 eligible participants, 132 participants were randomly assigned to participate in the fMRI sub-study. The Soroka Medical Center Medical Ethics Board and Institutional Review Board provided ethics approval. All participants provided written consent and received no financial compensation.

## Randomization and intervention

All participants completed the baseline measurements and were randomized, using a computer-based program, in a 1:1:1 ratio, stratified by sex and work status (to ensure equal workplace-related lifestyle features between groups), into one of the three intervention groups: healthy dietary guidelines (HDG) as an active control group, Mediterranean diet (MED), green-MED, all combined with PA. Interventions lasted for 18 months and were contemporaneous, and participants were not blind to group assignment (open-label protocol). Each participant received complete dietary guidance (based on the specific intervention group) and a free and fully available clinical dietitians consult. Furthermore, all participants received free gym membership, including educational sessions encouraging moderate-intensity PA. Participants in both MED groups were assigned to a diet rich in vegetables, with poultry and fish replacing beef and lamb, with 1500–1800 kcal/day for men, 1200–1400 kcal/day for women. The diet additionally included 28 g/day of walnuts (+440 mg/day polyphenols provided). The green-MED group further consumed green tea (3–4 cups/day) and Wolffia globosa green shake (100 g/day frozen cubes, +1240 mg/day total polyphenols provided). A detailed description of the intervention outline is available in *Supplementary file 1*.

## MRI acquisition

MRI scans were conducted at the Soroka University Medical Center (SUMC), Beer Sheva. Participants were scanned in a 3T Philips Ingenia scanner (Amsterdam, The Netherlands) equipped with a standard head coil. Subjects were instructed to refrain from food and non-water beverages 2 hr before the MRI sessions. Each of the two sessions at T0 and T18 included 2 RS-fMRI runs of 7 min each and a 3D T1-weighted anatomical scan to allow registration of the functional data. Before each RS session, participants were instructed to remain awake with their eyes open and lie still. fMRI BOLD contrast was acquired using the gradient-echo echo-planner imaging sequence with parallel acquisition (SENSE: factor 2.2). Scanning parameters were as follows: whole-brain coverage 41 slices ($3 \times 3 \times 3$ mm$^3$), transverse orientation, 3 mm thickness, no gap, TR = 2200 ms, TE = 30 ms, flip angle = 90°, FOV = 200 × 222 (RL × AP) and matrix size 68 × 71 (RL × AP). High-resolution anatomical volumes were acquired with a T1-weighted 3D pulse sequence ($1 \times 1 \times 1$ mm$^3$, 150 slices).

## MRI preprocessing

The preprocessing pipelines used in this work were extensively described in a previous publication (*Levakov et al., 2021*). T1w scans were preprocessed through FreeSurfer's (*Fischl et al., 1999*)

(version 6.0) recon-all processing. FreeSurfer's cortical segmentation and spherical warp were used to transfer the Schaefer 100-node cortical parcellation (*Schaefer et al., 2018*) to each subject's volumetric anatomical space. Functional images of the NKI dataset were preprocessed with fMRIPrep (version 1.1.8; *Esteban et al., 2019*) and images of the DIRECT-PLUS and Cam-CAN datasets were preprocessed with the Configurable Pipeline for the Analysis of Connectomes (C-PAC [*Cameron et al., 2013*] version 1.6.2). Briefly, both pipelines included the following steps: slice-timing correction, motion correction, skull stripping, estimation of motion parameters, and other nuisance signal time series. For the NKI dataset, functional scans were bandpass filtered (0.008–0.08 Hz) and confound regressed in a manner orthogonal to the temporal filters. Confounds included six motion estimates, the mean time series derived in CSF, WM, and whole-brain masks, the derivatives of these nine regressors, and the squares of these 18 terms. Spike regressors were added for each frame with framewise displacement above 0.5 mm. Data were linearly detrended and standardized. Nuisance regression in the DIRECT-PLUS and Cam-CAN fMRI dataset included the first five principal components of the signal from white matter and CSF (*Behzadi et al., 2007*), six motion parameters, and linear and quadratic trends, global signal regression, followed by temporal filtering between 0.1 and 0.01 Hz. Finally, a scrubbing threshold of 0.5 mm frame-wise displacement was applied (*Power et al., 2014*) (removal of 1 TR before and 2 TR after excessive movement). The time series of the two functional scans in the DIRECT-PLUS were concatenated to a single T0 and T18 scans. The exclusion criterion for excessive movements was determined a priori to less than 70% (9 min and 48 s) of the resting-state session after the scrubbing procedure (23% omitted; 102 subjects left). In all datasets, functional connectivity was defined as the Pearson's correlation among pairs of ROIs' time series followed by Fisher's r-to-z transformation.

## Clinical measurement and blood biomarkers

All parameters were measured at baseline and after 18 months of intervention. All clinical measures in the current study were selected a priori from a large pool of variables taken in the DIRECT-PLUS trial (*Yaskolka Meir et al., 2021b*). These measures were taken from five pre-selected categories: (1) Anthropometry that includes body mass index (BMI) and WC; (2) liver biomarkers that included AST, alanine transaminase (ALT), gamma-glutamyl transferase (GGT), ALKP, FGF 21, and chemerin; (3) glycemic markers, including glucose HOMA-IR and HbA1c; (4) lipids including cholesterol, HDL-C, LDL-C, and triglycerides; (5) imaging measures included liver fat, VAT, DSC, SSC, and the hippocampal occupancy score (HOC). WC was measured to the nearest millimeter halfway between the last rib and the iliac crest using an anthropometric measuring tape. Blood and urine samples were collected at 8:00 AM after a 12 hr fast. Blood samples were centrifuged and stored at –80°C. HOC was calculated as the hippocampal volume divided by (hippocampal volume + inferior lateral ventricle volume) in each hemisphere, then averaged across hemispheres (*Kaplan et al., 2022*; *Heister et al., 2011*).

## Nutritional assessment

Assessment of nutritional intake and lifestyle habits was self-reported online using validated food frequency questionnaires (FFQ) (*Shai et al., 2005*). The questionnaires were administered at baseline, after 6 months, and at the end of the trial. We selected a priori the questionnaire variables that were associated with brain age attenuation. These variables included the change in the following categories: sweets and beverages, weekly *Wolffia globose* intake, nut and seeds, eggs and milk, beef, processed food, green tea, and walnuts. The closed workplace enabled monitoring of the freely provided lunch and the intense dietary and PA sessions, which were provided simultaneously to all three groups.

## Liver and visceral fat imaging protocols

To quantify and follow IHF% changes, we used H-MRS, a reliable tool for liver fat quantification (*Kukuk et al., 2015*). Localized, single-voxel proton spectra were acquired using a 3.0 T magnetic resonance scanner (Philips Ingenia, Best, The Netherlands). The measurements were taken from the right frontal lobe of the liver, with a location determined individually for each subject using a surface, receive-only phased-array coil. Spectra with and without water suppression were acquired using the single-voxel stimulated echo acquisition mode with the following parameters: TR = 4000 ms, TE = 9.0 ms, and TM = 16.0 ms. The receiver bandwidth was 2000 Hz and the number of data points was

1024. Second-order shimming was used. Four averages were taken in a single breath hold for an acquisition time of 16 s. The total image hepatic fat fraction was determined as the ratio between the sum of the area under all fat divided by the sum of area under all fat and water peaks (*Hu et al., 2010*).

Abdominal fat depots were assessed at baseline and 18 months thereafter using 3 T MRI scans (Ingenia 3.0T, Philips Healthcare, Best, the Netherlands). The scanner utilized a 3D modified DIXON imaging technique without gaps (2 mm thickness and 2 mm of spacing), fast-low-angle shot sequence with a multi-echo two excitation pulse sequence for phase-sensitive encoding of fat and water signals (TR, 3.6 ms; TE1, 1.19 ms; TE2, 2.3 ms; FOV 520 × 440 × 80 mm³; 2 × 1.4 × 1 mm³ voxel size). Four images of phantoms were generated: in-phase, out-phase, fat, and water phase (*Thomas et al., 2013*). Participants were instructed to hold their breath to avoid motion artifacts when their abdomen was scanned. A continuous line over the fascia superficialis was drawn to differentiate deep-SAT from superficial-SAT and calculated mean VAT, deep-SAT, and superficial-SAT along two axial slices: L5-S1 and L4-L5. We quantified fat mass regions as area and relative proportion of each fat subtype (percentage).

## Brain age estimation

Subjects' chronological age was predicted from the lower triangle of the functional connectivity matrix depicting all unique edges (4950 edges). We used a support vector regression model (*Smola et al., 2000*) implemented using Scikit-learn (*Fabian et al., 2011*) with a linear kernel and all the default parameters. Model accuracy was quantified as the Pearson's correlation between the observed and predicted age. We additionally report the mean absolute error (MAE) in years, along with a p-value based on a non-parametric permutation test created by shuffling the data labels 1000 times (*Hilger et al., 2020*).

## Statistical analysis

The primary outcome of the current work was brain age attenuation quantified as the difference between the expected and observed brain age at T18 (*Yaskolka Meir et al., 2021a*). The expected brain age at T18 was calculated by first producing brain age prediction for all participants at T0. Then, a linear regression was used to estimate brain age from the chronological age at T0. The fitted regression formula, representing the expected aging trajectory in the absence of intervention, was used to estimate the expected brain age at T18 given each participant's T0 brain age and the time passed between the T0 and T18 MRI scans. The observed brain age was produced by applying the brain age model to the T18 scans. At baseline, brain age gap was computed as the difference between the predicted and observed age after regressing out the effect of the chronological age on the this gap (brain age bias correction [*Smith et al., 2019*]). We note that the result of computing the difference between the bias corrected brain age gap at both times was nearly identical to the brain age attenuation measure ($r$=0.99, p<0.001; MAE = 0.45). The difference between the two is because the brain age attenuation model takes into account the difference in the exact time that passed between the two scans for each participant (mean = 21.36 m, std = 1.68 m). Association between brain age attenuation and change in clinical measures following the intervention were reported using Pearson's correlation. Correction for multiple comparisons was conducted within each biomarker category using the Benjamini–Hochberg false discovery rate (FDR; *Benjamini and Hochberg, 1995*) with an alpha of 0.05. Associations to food consumption reports were reported using Kendall's tau correlation for ordinal data. Processed food at T18 had only two levels, 'same consumption' and 'less consumption', thus relation to brain age attenuation was tested with independent t-test. Change in clinical measurements were computed as a delta (Δ), the value at T0 minus the value at T18. We quantified change in reported food consumption as the change between the T0 and T18 questionnaires for food groups (i.e. processed food, sweets, and beverages) and as total consumption for polyphenols-provided foods (i.e. Mankai, green tea, walnuts). To control for the possible effect of age or gender, we used partial regression by regressing out the linear effect of age and gender from both brain age attenuation and the clinical measures. This was done by predicting each clinical measure, with the covariate as a predictor, keeping only the residual.

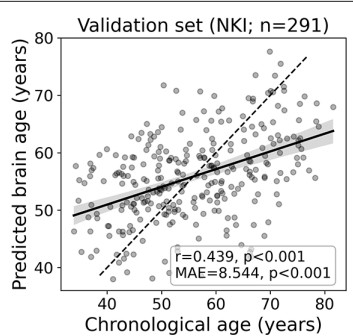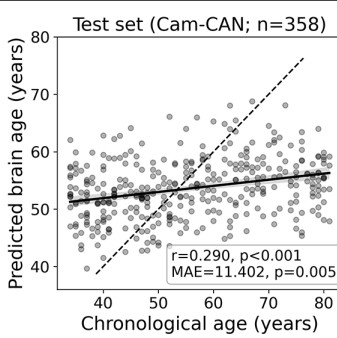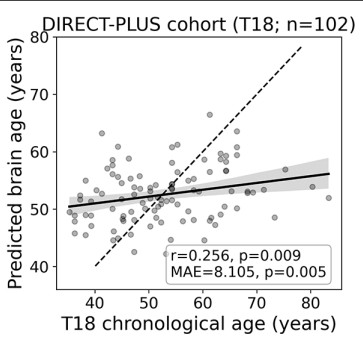

**Figure 2.** Prediction accuracy within the validation and test cohorts. The scatter plots depict the data points and regression line between the predicted (y-axis) and observed (x-axis) age. The predicted-observed correlation is presented for the validation data (left), the Cambridge Centre for Ageing and Neuroscience (Cam-CAN) test data (middle), and the Dietary Intervention Randomized Controlled Trial Polyphenols Unprocessed Study (DIRECT-PLUS) data at baseline. The shaded area around the regression lines represents a 95% confidence interval estimated using bootstrapping. Pearson's correlation, MAE (mean absolute error), and corresponding p-values are shown at the bottom of each plot. The dotted lines represent a perfect correlation for reference (predicted = observed).

The online version of this article includes the following figure supplement(s) for figure 2:

**Figure supplement 1.** Brain age prediction accuracy of individual nodes.

## Effect size in previous work

In a recent study (*Zeighami et al., 2022*), the authors reported a significant decrease in delta age 12 months following bariatric surgery. We converted the reported t statistic (t=3.66, p<0.001, df = 85) to an effect size (Cohen's d=0.79, *r*=0.369) using an effect size calculator (https://lbecker.uccs.edu/). Using a sample size calculator (Python statsmodels.stats.power), we found that with an alpha of 0.05 and beta >0.05, a sample size >90 is required. With the given sample size (n=102), the probability of failing to reject the null hypothesis under the alternative hypothesis (β, Type II error rate) is 3%. Alternatively, for a Type II error rate lower than 0.1 with the given sample size, an effect size of 0.664 (Cohen's d) is required.

## Results

### Brain age estimation

To estimate chronological age from RSFC, we utilized data from 649 participants from two separate cohorts for the brain age model training, validation, and testing. We predicted chronological age from functional connectivity among the 100 nodes of the Schaefer brain atlas (*Schaefer et al., 2018*) (4950 edges) using a linear support vector regression model. The model was first trained and validated on 291 participants from the NKI dataset (*Nooner et al., 2012*; n=291) using fivefold cross-validation. As expected, a positive correlation was found between the predicted and observed age (*r*=0.439, p<0.001; MAE = 8.544, p<0.001). Next, we retrained the model on the entire sample and tested it in an independent sample from the Cam-CAN dataset (*Shafto et al., 2014*; n=358) again, yielding a positive correlation between the predicted and observed age (*r*=0.290, p<0.001; MAE = 11.402, p=0.005). Finally, we used the fitted model to estimate the brain age within the DIRECT-PLUS cohort. Of the 132 subjects that participated in the fMRI sub-study, 102 were included in all analyses after exclusions due to excessive in-scanner motion (23% omitted; MRI preprocessing). The predicted brain age and observed chronological age were correlated (*r*=0.244, p=0.013; MAE = 8.337, p<0.001; *Figure 2*), reproducing the results found within the two other datasets. Despite being significant and reproducible, we note that the correlations between the observed and predicted age were relatively moderate.

### Baseline characteristics

Baseline characteristics among the 102 participants with valid RSFC MRI scans are presented in *Table 1* (see *Supplementary file 2* for additional measures). The mean participant age was 51.5±10.5 years (median = 50.6, range 33.9–81.9), and 91.2% were men. The mean BMI and WC were 30.1±2.5 kg/m²

**Table 1.** Association between baseline characteristics and age, predicted age, and brain age gap.

| | Age | | Brain age | | T0 brain age gap (bias corrected) | |
|---|---|---|---|---|---|---|
| | r | p-Value | r | p-Value | r | p-Value |
| BMI (kg/m²) | –0.110 | 0.272 | 0.067 | 0.504 | 0.097 | 0.334 |
| Chemerin (ng/mL) | 0.153 | 0.124 | **0.247** | **0.012** | **0.216** | **0.029** |
| HOMA IR | 0.177 | 0.079 | 0.179 | 0.074 | 0.141 | 0.163 |
| HbA1c (%) | 0.315 | 0.001 | 0.117 | 0.240 | 0.042 | 0.677 |
| HDL-C (mg/dL) | 0.158 | 0.113 | –0.095 | 0.343 | –0.138 | 0.168 |
| LDL-C (mg/dL) | –0.152 | 0.129 | –0.024 | 0.811 | 0.013 | 0.893 |
| Triglycerides (mg/dL) | –0.023 | 0.815 | 0.145 | 0.147 | 0.155 | 0.120 |
| Liver fat (cm²) | –0.039 | 0.711 | 0.156 | 0.132 | 0.170 | 0.101 |
| VAT (cm²) | **0.495** | **0.000** | **0.329** | **0.001** | **0.225** | **0.025** |

and 107.1±6.6 cm, respectively. The mean baseline predicted brain age by RSFC was 52.8±4 years. In *Table 1* we report the correlation between baseline characteristics and age, predicted age, and the brain age gap. The brain age gap was computed as the difference between the predicted and observed age after regressing out the effect of age on the this gap (brain age bias correction [*Smith et al., 2019*]). At baseline, brain age gap was correlated with chemerin ($r=0.22$, p=0.029), and with obesity-associated measurements obtained by MRI including visceral abdominal tissue (VAT): $r=0.23$, p=0.02 and superficial subcutaneous fat (SSC): $r=-0.25$, p=0.014.

## The relation between successful lifestyle intervention and attenuation of functional brain aging

Our primary hypothesis was that success in lifestyle intervention, as assessed by anthropometric measurements, will attenuate functional brain aging. Brain aging attenuation was quantified as the difference between the expected and observed brain age at T18 (*Yaskolka Meir et al., 2021a*; *Figure 1e*). Following 18 months of lifestyle intervention, participants showed a reduction of 0.76 (±1.86) units in BMI on average, 2.31 (±5.61) kg reduction in weight, and 5.39 (±5.89) cm reduction in WC. These constitute a $-6.45\% \pm (5.60\%)$ and $-4.35\% \pm (5.86\%)$ reduction from baseline for WC and BMI and weight, respectively. Additionally, at T18, the observed age was lower than expected in 56.8% of the subjects, while the opposite was found in 43.1% of the subjects ($X^2=1.922$, p=0.166; see *Figure 3*, top). Importantly, we found a correlation between ΔBMI and brain age attenuation such that participants that showed a decrease in BMI also exhibited attenuated brain aging ($r=0.319$, p<0.001; *Figure 3*, bottom). Specifically, 1% of BMI or weight loss resulted in an 8.9 months' attenuation of brain age (*Figure 3—figure supplement 1*). Similar results were found with Δbody weight ($r=0.319$, p<0.001) and ΔWC ($r=0.198$, p=0.046; *Figure 4*). The correlations to ΔBMI and Δweight were significant after correcting for age and baseline brain age (p<0.05 for all), while the correlation to WC did not show a significant association ($r>0.171$, p's <0.079 for all).

## The relation between brain age attenuation and clinical measurements

To examine the clinical outcomes associated with attenuated brain aging, we further tested the correlation of brain age attenuation with liver, glycemic, lipids, and MRI-assessed fat deposition biomarkers (*Figure 4*). Except of deep subcutaneous changes, all fat deposition measurements, superficial subcutaneous, visceral, and liver fat changes (e.g. loss) were significantly and directly associated with brain age attenuation (p<0.05, FDR corrected), i.e., the more the individual succeeded in diet-induced fat depots loss, the more brain age attenuation has been achieved. We then tested the association between brain age attenuation and liver and glycemic biomarkers. Out of all examined liver biomarkers, a decrease in ALT, GGT, alkaline phosphatase, and serum chemerin were significantly associated with attenuation in brain age (p<0.05 for all, FDR corrected). Of all examined lipid profile markers, only an increase in ΔHDL-C was significantly correlated with

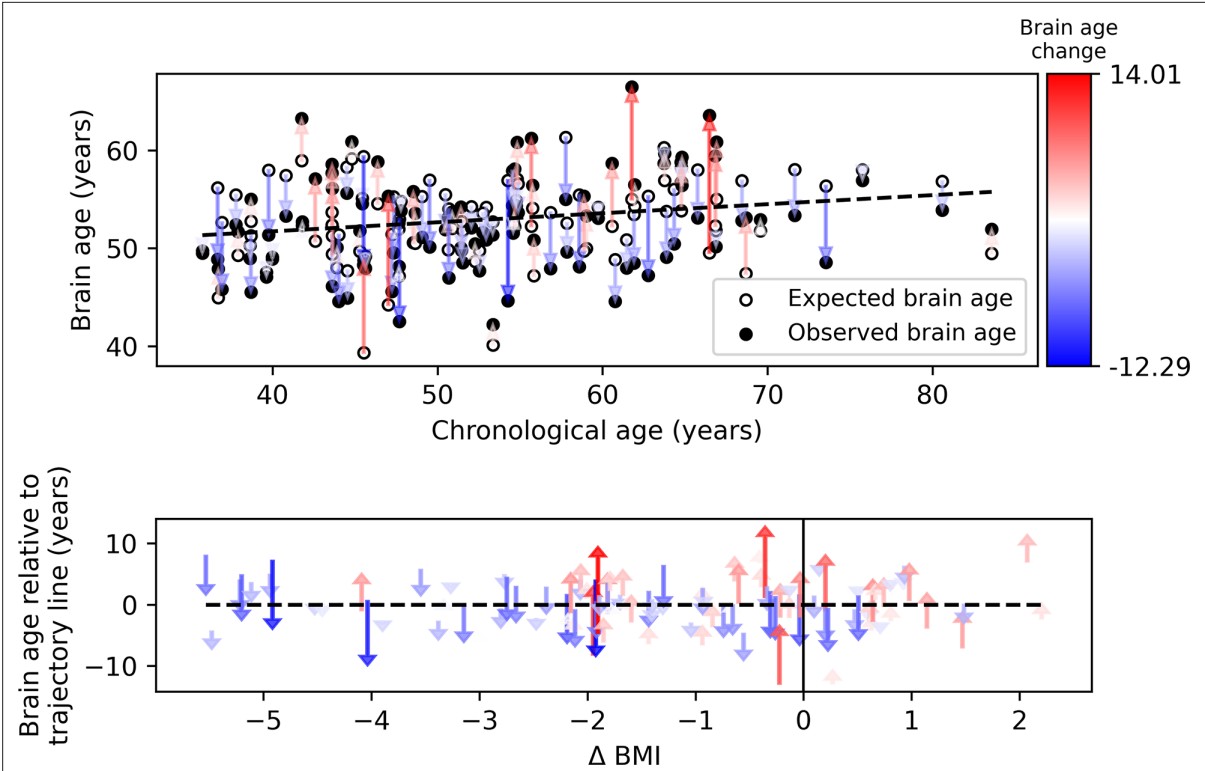

**Figure 3.** Observed compared to expected brain age at T18. The upper panel depicts the chronological age (x-axis) and the observed (empty circles) and expected (full circles) brain age (y-axis) of each subject. The dashed line represents the expected brain age trajectory fitted based on the T0 data (see regression line in *Figure 1e*, left). Arrows point from the expected to the observed age of a single individual, corresponding to brain age attenuation. Arrows' colors correspond to the extent of brain age attenuation (blue shades indicate attenuation, red shades indicate an acceleration in brain age). The observed age was lower than expected in 56.8% of the subjects, while the opposite was found in 43.1% ($X^2$=1.922, p=0.166). In the lower panel, arrows were reordered by subjects' body mass index (BMI) change over the 18 months of intervention. A significant correlation was found between the BMI and brain age change (r=0.319, p<0.001). This is evident in the graph, such that most of the blue arrows are located on the left side of the x-axis (negative values), and most of the red arrows appear on the right side (positive values).

The online version of this article includes the following source data and figure supplement(s) for figure 3:

**Source data 1.** Participants' demographics predicted and observed age and weight values.

**Figure supplement 1.** Brain age attenuation compared to percent weight reduction from baseline.

brain age attenuation (r=-0.273, p=0.005). Finally, a decrease in HOC was significantly correlated with brain age attenuation (r=-0.296, p=0.003). All results were reproduced after controlling for baseline age and predicted age at T0. However, after further correction for changes in BMI, only Δalkaline phosphatase, Δchemerin, and ΔHOC were associated with brain age attenuation (all p's<0.018), with no significant associations with all other biomarkers (p>0.05, for all; see *Supplementary file 3*).

## The relation between brain age attenuation and food consumption

We examined whether food consumption, as reported using an FFQ, could be associated with functional brain aging attenuation. Associations were tested using Kendall's rank correlation. We began with categories that could negatively affect brain aging attenuation. In line with our hypothesis, we found that decreased consumption of processed food (t=3.131, p=0.002) and sweets and beverages ($\tau$=−0.231, p=0.002) was associated with more attenuation in brain age. An increase in green tea and walnut consumption, for which we hypothesized an attenuation effect on brain aging due to their high polyphenol content, did not result in a significant correlation (all $\tau$'s<0.081, p's>0.121; see *Supplementary file 4* for all measures).

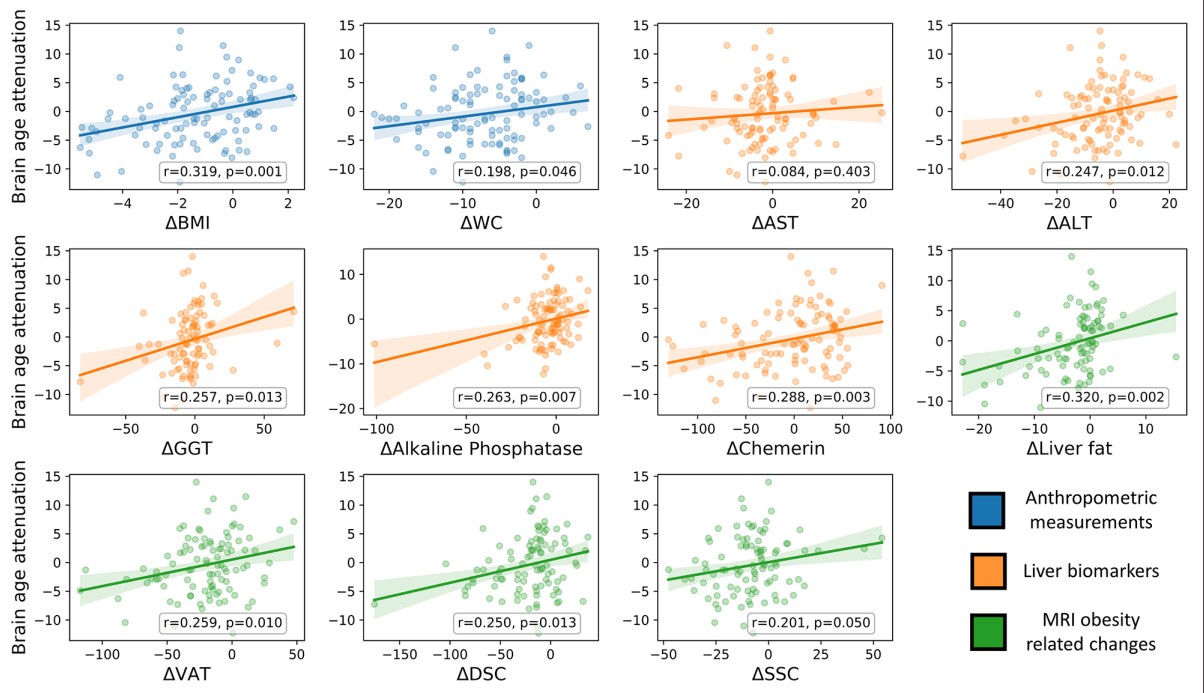

**Figure 4.** Brain age attenuation association with clinical measurements. The scatter plots depict the data points and regression line between brain age attenuation (y-axis) and each clinical measurement (x-axis). Clinical measurements include anthropometry (blue), liver markers (orange), glycemic markers (brown), lipid profile (red), and fat deposition measured using magnetic resonance imaging (MRI) (green). The shaded area around the regression line represents a 95% confidence interval estimated using bootstrapping. Pearson's correlation and the corresponding p-value are shown at the bottom of each plot. Significant associations following false discovery rate (FDR) correction are marked in bold (*=p < 0.05, **=p < 0.01).

## Discussion

Considerable evidence implies that excessive weight accelerates normal aging (*Salvestrini et al., 2019*; *Tam et al., 2020*), a process that is also manifested in brain aging (*Beck et al., 2022*). In the current study, we examined for the first time whether weight loss following a lifestyle dietary intervention may attenuate the effect of obesity on the brain aging trajectory. We hypothesized that reducing anthropometric measurements following a lifestyle intervention would be associated with attenuated brain aging. We first demonstrated, across two separate cohorts, that age could be estimated from RSFC, as done in previous work (*Dosenbach et al., 2010*). We then applied the fitted age prediction model to the participants of the DIRECT-PLUS. We found that 1% of body weight loss results in an 8.9 months' attenuation of brain age. Attenuated brain aging was further correlated with a decrease in WC, MRI-assessed fat deposition, liver biomarkers, and HDL-C. Finally, reduced reported consumption of processed food, sweets and beverages were also related to attenuated brain aging. During the 18-month trial, a modest average weight loss of 2.31 (±5.61) kg was observed. As demonstrated by our previous trials, as well as others, the maximum weight loss following lifestyle intervention is achieved in 6 months, following a regain/rebound phase. In addition, our trials and others have demonstrated an improvement in cardiometabolic health following a MED intervention (i.e. hepatic and visceral fat reduction and improved lipid profile), despite a modest weight loss (*Yaskolka Meir et al., 2021b*; *Zelicha et al., 2022*; *Gepner et al., 2018*; *Estruch et al., 2013*).

Accumulated evidence points to the potential of lifestyle intervention to reverse the negative impact of excess weight on brain structure, function, and cognition. Cross-sectional and longitudinal studies found that reported adherence to a MED was linked to increased gray matter volume in multiple regions (*Staubo et al., 2017*), including the hippocampus (*Ballarini et al., 2021*). Adherence to HDG was also associated with reduced cognitive decline (*Gardener and Rainey-Smith, 2018*). Importantly, randomized clinical trials can support a causal relationship between lifestyle intervention and the brain aging process. Such studies from our group (*Kaplan et al., 2022*) and others (*Erickson et al., 2011*; *Espeland et al., 2016*) revealed that subjects enrolled in a PA+dietary intervention exhibit

lower hippocampal atrophy and smaller ventricles. A similar beneficial effect on cognitive functioning in middle age was also found (*Valls-Pedret et al., 2015*), along with functional connectivity alteration in the default mode and executive control networks (*Voss et al., 2010*; *García-Casares et al., 2017*). To date, a single study in rats has tested the effect of a dietary intervention using the brain age framework and found a reduction in brain aging rate (*Brusini et al., 2022*). Hence, the current work provides the first evidence that such a beneficial effect on brain age can also be found in humans.

Studying changes in functional brain aging is part of a broader field that examines changes in various biological ages, such as telomere length (*Gampawar et al., 2020*), DNA methylation (*Fraga, 2009*), and arterial stiffness (*Hamczyk et al., 2020*). Evaluating changes in these bodily systems over time allows us to capture health and lifestyle-related factors that affect overall aging and may guide the development of targeted interventions to reduce aging-related decline. For example, in the CENTRAL cohort, we recently reported that reducing body weight and intrahepatic fat following a lifestyle intervention was related to methylation age attenuation (*Yaskolka Meir et al., 2021a*). In the current work, we used RSFC for brain age estimation, which resulted in an MAE of ~8 years, that was larger than the intervention period. Nevertheless, we found that brain age attenuation was associated with changes in multiple health factors. The precision of an age prediction model based on RSFC is typically lower than a model based on structural brain imaging (*de Lange et al., 2022*). However, a higher model precision may result in a lower sensitivity to detect clinical effects (*Bashyam et al., 2020*; *Jirsaraie et al., 2023*). Better tools for data harmonization among datasets (*Jirsaraie et al., 2023*) and larger training sample size (*de Lange et al., 2022*) may improve the accuracy of such models in the future. We also suggest that examining the dynamics of multiple bodily ages and their interactions would enhance our understanding of the complex aging process (*Yu et al., 2022*; *Franke and Gaser, 2019*).

The brain age framework reduces the multifaceted aging process captured in a given imaging modality to a single scalar. This scalar, the predicted brain age, is well defined in the sense that it minimizes the prediction error within the training dataset. Moreover, the clinical relevance of functional brain age is shown, for example, in predicting Alzheimer's onset (*Gonneaud et al., 2021*) and symptoms severity in depression (*Dunlop et al., 2021*). This reductionist approach raises several challenges. The first is the ability to interpret the features used by the machine learning model (*Levakov et al., 2020*). A second challenge is understanding the physiological factors that may affect its predictions (*Mora, 2013*), which we address in the current work. Here, we report how a set of clinical measures are associated with changes in brain aging. Importantly, lifestyle and other interventions can affect these measures to attenuate the brain aging process. We suggest that such mapping of changes in clinical outcomes to months or years of attenuated brain aging has important scientific, clinical, and even educational value.

We found that clinical outcomes that include anthropometric, liver, and lipid biomarkers were associated with attenuated brain age. Specifically, two main factors were linked to changes in brain age, changes in anthropometry measures, and liver status. The first factor included BMI, weight, WC, and superficial subcutaneous and visceral fat. The second factor included liver fat, ALT, GGT, alkaline phosphatase, and serum chemerin. Alkaline phosphatase and chemerin were also associated with changes in brain age after controlling for changes in BMI. The negative impact of elevated liver enzymes and liver fat on brain health is seen, for example, in the case of AD (*Nho et al., 2019*; *Labenz et al., 2021*; *Ghareeb et al., 2011*). This link is thought to be mediated by oxidative stress, vascular damage, and inflammation (*Helfer and Wu, 2018*). Chemerin, produced in the liver, is an adipokine linked to energy homeostasis, adipogenesis, and excessive weight (*Helfer and Wu, 2018*). Chemerin is correlated with age (*Aronis et al., 2014*) and BMI (*Ernst and Sinal, 2010*) and was found to be reduced following lifestyle intervention (*Blüher et al., 2012*; *Ashtary-Larky et al., 2021*). The relation between serum chemerin and brain aging is still unclear, but possible linking mechanisms are hypertension (*Ferland et al., 2020*) and inflammation (*Ernst and Sinal, 2010*). Besides these two factors, HDL-C was the only variable whose increase was correlated to brain aging attenuation. This is in line with evidence of the protective role of HDL-C in cognitive decline and dementia (*Hottman et al., 2014*). Finally, of all the reported food consumption items, only reduced consumption of processed food, sweets and beverages were linked to attenuated brain aging. Although these results are based on self-reports, they may be helpful for developing neuroprotective dietary guidelines (*Smith et al., 2010*).

It is important to consider several limitations and strengths of the current study. The first limitation was gender imbalance (F: 93, M: 9; F: 8.8%, M: 91.2%), which reflected the workplace profile from which participants were recruited (*Yaskolka Meir et al., 2019*; *Tsaban et al., 2021*; *Rinott et al., 2021*; *Zelicha et al., 2019*). This distribution misrepresents the proportion of obese women within the general population (F: 51%, M: 49%; *Craig et al., 2020*). Hence, these results should be further corroborated in a gender-balanced sample. Additionally, participants were recruited based on excess adiposity or dyslipidemia, therefore, they represented a restricted range of the normal population. This design allows to maximize the intervention effects but restricts our ability to detect correlation at baseline. We also note that the lack of a no-intervention control group limits our ability to directly relate our findings to the intervention. Hence, we can only relate brain age attenuation to the observed changes in health biomarkers. The strengths of the study lies in the wealth of health biomarkers that included anthropometric, blood, and imaging measures, the relatively large sample for similar intervention trials, the tight on-site monitoring over the dietary compliance, and the long intervention duration. Finally, the use of three distinct datasets for training and validation, testing, and inference supports the generalization of our model.

To conclude, in the current work, we examined how changes in multiple health factors, including anthropometric measurements, blood biomarkers, and fat deposition, can account for brain aging attenuation. We reveal that the two factors with the strongest association with brain aging were changes in anthropometry measures and liver biomarkers. These findings complement the growing interest in bodily aging indicated, for example, by DNA methylation (*Yaskolka Meir et al., 2021a*) as health biomarkers and interventions that may affect them. These exciting results may advance our knowledge of factors related to healthy brain aging and guide future neuroprotective interventions.

## Acknowledgements

This work was supported by grants from: the German Research Foundation (DFG), German Research Foundation - project number 209933838 - SFB 1052; B11, Israel Ministry of Health grant 87472511 (to I Shai); Israel Ministry of Science and Technology grant 3-13604 (to I Shai); and the California Walnuts Commission (to I Shai).

## Additional information

### Competing interests

Matthias Blüher: has received consulting fees from Amgen, Astra Zeneca, Boehringer-Ingelheim, Bayer, Lilly, Novo Nordisk, Novartis, Sanofi and Pfizer; and fees for lectures/ presentations from Amgen, Astra Zeneca, Boehringer-Ingelheim, Bayer, Daiichi-Sankyo, Lilly, Novo Nordisk, Novartis, Sanofi and Pfizer. The author is also on the advisory board for Boehringer-Ingelheim. The author has no other competing interests to declare. The other authors declare that no competing interests exist.

### Funding

| Funder | Grant reference number | Author |
| --- | --- | --- |
| The German Research Foundation | 209933838 SFB 1052 | Iris Shai |
| Israel Ministry of Health | grant 87472511 | Iris Shai |
| Israel Ministry of Science and Technology | 3-13604 | Iris Shai |
| California Walnut Commission | 09933838 SFB 105 | Iris Shai |
| The German Research Foundation | B11 | Iris Shai |

The funders had no role in study design, data collection and interpretation, or the decision to submit the work for publication.

## Author contributions
Gidon Levakov, Conceptualization, Software, Formal analysis, Validation, Investigation, Visualization, Methodology, Writing – original draft, Writing – review and editing; Alon Kaplan, Data curation, Formal analysis, Validation, Investigation, Methodology, Writing – original draft, Project administration, Writing – review and editing; Anat Yaskolka Meir, Resources, Data curation, Project administration, Writing – review and editing; Ehud Rinott, Gal Tsaban, Hila Zelicha, Resources, Data curation, Project administration; Matthias Blüher, Uta Ceglarek, Michael Stumvoll, Resources, Project administration; Ilan Shelef, Resources, Methodology, Writing – review and editing; Galia Avidan, Supervision, Investigation, Methodology, Project administration, Writing – review and editing; Iris Shai, Conceptualization, Resources, Data curation, Supervision, Funding acquisition, Validation, Investigation, Project administration, Writing – review and editing

## Author ORCIDs
Gidon Levakov  http://orcid.org/0000-0002-5520-3556
Alon Kaplan  http://orcid.org/0000-0002-9123-8094
Galia Avidan  http://orcid.org/0000-0003-2293-3859

## Ethics
Clinical trial registration clinicaltrials.gov ID: NCT03020186.
This work was based on a sub-study of the DIREC-PLUS trial (clinicaltrials.gov ID: NCT03020186). The Soroka Medical Center Medical Ethics Board and Institutional Review Board provided ethics approval. All participants provided written consent and received no financial compensation.

## Decision letter and Author response
Decision letter https://doi.org/10.7554/eLife.83604.sa1
Author response https://doi.org/10.7554/eLife.83604.sa2

---

# Additional files

### Supplementary files
• Supplementary file 1. Intervention outline by group. A table describing the details of the intervention for each of the three groups.

• Supplementary file 2. Association between baseline characteristics and age, predicted age, and brain age gap. A table describing the Pearson's correlation between baseline characteristics and age, predicted age, and brain age gap.

• Supplementary file 3. Correlation and partial correlation of brain age attenuation and the measured biomarkers. A table describing the Pearson's correlation and partial correlation of brain age attenuation and the change in the measured biomarkers.

• Supplementary file 4. Relation between brain age attenuation and food consumption. A table describing the Kendall's tau correlation coefficient of brain age attenuation and food consumption, as reported using a food frequency questionnaire (FFQ).

• MDAR checklist

### Data availability
The code for the brain age prediction model and the calculation of brain age attenuation is openly available online at https://github.com/GidLev/functional_brain_aging (copy archived at *Levakov et al., 2023*). The unprocessed data used for the model training and validation is openly available online at http://fcon_1000.projects.nitrc.org/indi/enhanced/neurodata.html for the eNKI dataset and available upon online access request https://camcan-archive.mrc-cbu.cam.ac.uk/dataaccess/ for the CamCAN dataset. Data from the DIRECT-PLUS trial is not publicly available since it contains information that could compromise the privacy of research participants. However, de-identified data could be shared upon request, subject to approval from the Soroka Medical Center Medical Ethics Board. A processed version of the data that includes participants' demographics, predicted and observed age and weight values for T0 and T18 is available as Figure 3—source data 1.

The following previously published datasets were used:

| Author(s) | Year | Dataset title | Dataset URL | Database and Identifier |
|---|---|---|---|---|
| Taylor JR, Williams N, Cusack R, Auer T, Shafto MA, Dixond M, Dixond LK, Tylerd LK, Cam Can, Richard NH | 2017 | Magnetoencephalography (MEG) | https://camcan-archive.mrc-cbu.cam.ac.uk/dataaccess/ | The Cambridge Centre for Ageing and Neuroscience, 10.48532/009000 |
| Nooner KB, Colcombe SJ, Tobe RH, Mennes M, Benedict MM, Moreno AL, Panek LJ, Brown S, Zavitz ST, Li Q, Sikka S | 2012 | NKI-RS Enhanced Sample | https://identifiers.org/RRID:SCR_010461 | International Neuroimaging Data-sharing Initiative, SCR_010461 |

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
