## [Editor Report]

This study is indeed a landmark work that reports the significant benefits of lifestyle intervention in terms of attenuation of brain age and improvement in several tissue-based biomarkers. The findings from this study are of compelling and convincing nature that would encourage and support structured lifestyle intervention as an inclusive part of public health.

---

## [Decision Letter]

**Decision letter after peer review:**

Thank you for submitting your article "The effect of 18 months lifestyle intervention on brain age assessed with resting-state functional connectivity" for consideration by *eLife*. Your article has been reviewed by 3 peer reviewers, including Muthuswamy Balasubramanyam as the Reviewing Editor and Reviewer #1, and the evaluation has been overseen by Christian Büchel as the Senior Editor. The following individual involved in the review of your submission has agreed to reveal their identity: Ann-Marie de Lange (Reviewer #2).

Essential revisions:*Reviewer 1: (Essential Revisions)*

Authors should refer to the articles by Gampawar et al. (2020); Yu et al. (Neurobiol Aging 2022) and Franke & Gaser (2019) and extend/substantiate their discussion.

*Reviewer 2: (Essential Revisions)*

Introduction: Query 1

The authors could consider an introductory paragraph explaining RSFC and why it was used in this study: how does RSFC link to lifestyle and health, and how does it change with aging (e.g., what characteristics are expected)?

Methods: Queries 1 & 2

1. Line 317: The authors could clarify their inclusion and exclusion criteria. For inclusion criteria, the authors could cite relevant papers defining the cut-off values as thresholds to obesity. For exclusion, are the cut-off values linked to e.g., worse body or brain outcomes, or why were they included? Relevant citations may also be beneficial here.

2. The authors could consider including key details on the intervention study (those listed in SI Table 3), which are integral to the interpretations, in the main body of the text, rather than supplementary analyses (e.g., kcal restrictions). It would also be helpful if the authors could clarify how the randomization to different intervention groups is accounted for in the current study.*Reviewer 3: (Essential Revisions)*

Answers to comments: 1, 3 & 4

1) The authors did not choose to calculate brain age gap estimates for T0 and T18, but instead projected the brain-predicted age from T0 to T18 and calculated the difference between this 'expected brain age' and the 'observed brain age'. This procedure appears to be rather unusual. Is there any citeable resource that inspired the authors to follow this approach?

2. Question related to: how the effect size of 0.79 was derived.

3. The study uses a variety of variables and it remains uncertain if all relevant variables have been mentioned. For instance, the different categories derived from the food frequency questionnaires have not been described sufficiently. How many categories have been tested for an association with changes in brain age? In order to increase transparency, I suggest providing the "21-word solution" by Simmons et al. (http://dx.doi.org/10.2139/ssrn.2160588): "We report how we determined our sample size, all data exclusions (if any), all manipulations, and all measures in the study." If necessary, this statement might be adjusted to ensure that it is accurate.

*Reviewer #1 (Recommendations for the authors):*

Authors should consider addressing the comments with additional experiments and revising the manuscript.

Comments:

1) Authors should refer to Yu et al. (Neurobiol Aging 2022) and extend their discussion.

2) Authors should refer to the article by Franke & Gaser (2019) and extend/substantiate their discussion.

*Reviewer #2 (Recommendations for the authors):*

Introduction

1. The authors could consider an introductory paragraph explaining RSFC and why it was used in this study: how does RSFC link to lifestyle and health, and how does it change with ageing (e.g., what characteristics are expected)?

Results

1. Line 149: For a very restrictive intervention of only 1400 daily kcal for women and 1600 daily kcal for men over a period of 18 months, an average weight loss of 2.3kg seems small, especially in obese subjects. Could this potentially stem from a failure to comply with the intervention, resulting in lower weight loss or weight fluctuations? It would be helpful if the authors could comment on this in their discussion.

Methods

1. Line 317: The authors could clarify their inclusion and exclusion criteria. For inclusion criteria, the authors could cite relevant papers defining the cut-off values as thresholds to obesity. For exclusion, are the cut-off values linked to e.g., worse body or brain outcomes, or why were they included? Relevant citations may also be beneficial here.

2. The authors could consider including key details on the intervention study (those listed in SI Table 3), which are integral to the interpretations, in the main body of the text, rather than supplementary analyses (e.g., kcal restrictions). It would also be helpful if the authors could clarify how the randomisation to different intervention groups is accounted for in the current study.

*Reviewer #3 (Recommendations for the authors):*

1) The authors did not choose to calculate brain age gap estimates for T0 and T18, but instead projected the brain-predicted age from T0 to T18 and calculated the difference between this 'expected brain age' and the 'observed brain age'. This procedure appears to be rather unusual. Is there any citeable resource that inspired the authors to follow this approach?

2) Brain age was associated with several measures at baseline (Table 2), including chronological age. Usually, researchers in this field report on associations of the brain age gap, bias-corrected for chronological age. In other words, they report on the incremental associations of brain age with other variables, beyond the effects of chronological age. Without correcting for chronological age, associations of brain age with other variables are strongly driven by age. On this account, I feel that the results provided in Table 2 are not very informative with respect to the validity of the brain age estimates. The motivation for showing these statistics (message to the reader), as well as the choice of showing tertiles remains unclear to me.

3) I appreciate that the authors report a power analysis, but with the given information, it is not very comprehensible how the effect size of 0.79 was derived. I assume that this effect size estimate refers to Cohen's d!? Was it calculated based on the t-values and degrees of freedom from the '24-month post-surgery visits compared to baseline' reported in the cited resource? In general, effect sizes observed in single studies do not constitute very reliable estimates. In my view, a better option would be a sensitivity power analysis, which tests the probability (1-β) of different hypothetical effect sizes (e.g., d ranging from 0 to 1.5) to reach significance (α = 0.05) in a sample of respective size (N = 102; Lakens 2022, https://doi.org/10.1525/collabra.33267, see Figure 9).

4) The study uses a variety of variables and it remains uncertain if all relevant variables have been mentioned. For instance, the different categories derived from the food frequency questionnaires have not been described sufficiently. How many categories have been tested for an association with changes in brain age? In order to increase transparency, I suggest providing the "21-word solution" by Simmons et al. (http://dx.doi.org/10.2139/ssrn.2160588): "We report how we determined our sample size, all data exclusions (if any), all manipulations, and all measures in the study." If necessary, this statement might be adjusted to ensure that it is accurate.

---

## [Author Response]

Essential revisions:Reviewer #1 (Recommendations for the authors):Authors should consider addressing the comments with additional experiments and revising the manuscript.Comments:1) Authors should refer to Yu et al. (Neurobiol Aging 2022) and extend their discussion.2) Authors should refer to the article by Franke & Gaser (2019) and extend/substantiate their discussion.

We now relate to both studies in the Discussion section. We refer to our response to the previous comment for the citation from the text.

Reviewer #2 (Recommendations for the authors):Introduction1. The authors could consider an introductory paragraph explaining RSFC and why it was used in this study: how does RSFC link to lifestyle and health, and how does it change with ageing (e.g., what characteristics are expected)?

We now include such a paragraph in the Introduction section:

“We previously found that weight loss, glycemic control, and lowering of blood pressure, as well as an increment in polyphenols-rich food, were associated with an attenuation in brain atrophy ^12^. Obesity is also manifested in age-related changes in the brain’s functional organization as assessed with resting-state functional connectivity (RSFC). These changes are dynamic^13^ and can be observed in short time scales^14^ and thus of relevance when studying lifestyle intervention. Studies have linked obesity with decreased connectivity within the default mode network^15,16^ and increased connectivity with the lateral orbitofrontal cortex^17^, which are also seen in normal aging^18,19^. Longitudinal trials have reported changes in these connectivity patterns following weight reduction^20,21^, indicating that they can be altered. However, findings regarding functional changes are less consistent than those related to anatomical changes due to the multiple measures^22^ and scales^23^ used to quantify RSFC. Hence, focusing on a single measure, the functional brain age, may better capture these complex changes and their relation to aging. "

Results1. Line 149: For a very restrictive intervention of only 1400 daily kcal for women and 1600 daily kcal for men over a period of 18 months, an average weight loss of 2.3kg seems small, especially in obese subjects. Could this potentially stem from a failure to comply with the intervention, resulting in lower weight loss or weight fluctuations? It would be helpful if the authors could comment on this in their discussion.

We thank the reviewer for this important question. As demonstrated by our previous trials as well as by other studies, the maximum weight loss following lifestyle intervention is achieved in six months, where a nadir in weight loss is observed following a regain/rebound phase ^24–28^. In addition, our trials as well as others have demonstrated an improvement in cardiometabolic health following a Mediterranean diet intervention, i.e., hepatic and visceral fat reduction and improved lipid profile even when only modest weight loss is achieved ^26,29,30^. Thus, this moderate weight loss does not reflect a failure to comply with the intervention. Instead, it reflects the natural course of weight dynamics following a dietary intervention. Following the reviewer's suggestion, we now include the following text in the Discussion section:

“During the 18-month trial, a modest average weight loss of 2.31 (±5.61) kg was observed. As demonstrated by our previous trials, as well as others, the maximum weight loss following lifestyle intervention is achieved in six months, following a regain/rebound phase. In addition, our trials and others have demonstrated an improvement in cardiometabolic health following a Mediterranean diet intervention (i.e., hepatic and visceral fat reduction and improved lipid profile), despite a modest weight loss^42,51–53^.”

Methods1. Line 317: The authors could clarify their inclusion and exclusion criteria. For inclusion criteria, the authors could cite relevant papers defining the cut-off values as thresholds to obesity. For exclusion, are the cut-off values linked to e.g., worse body or brain outcomes, or why were they included? Relevant citations may also be beneficial here.

We thank the reviewer for this question. For the inclusion criteria, the cut-off values of waist circumference (WC): men>102 cm, women>88 cm were adopted from the NIH (https://www.nhlbi.nih.gov/sites/default/files/media/docs/obesity-evidence-review.pdf), and the CDC (cdc.gov/healthyweight/assessing/index.html), and are widely accepted in the literature. (i.e. ^31^). The cut-off values for triglycerides and HDL were adopted from the American Heart Association guidelines for metabolic syndrome ^32^. For exclusion criteria, we intended to recruit participants with obesity or dyslipidemia who were otherwise healthy. Thus, participants who were chronically ill (a significant illness that might require hospitalization; active cancer or chemotherapy treatment in the last three years; serum creatinine ≥2 mg/dL) or participants who could not undergo lifestyle or dietary intervention or an MRI scan were also excluded (inability to perform the physical activity; pregnancy or lactation; warfarin treatment due to possible interaction with Wolffia globose which has a high vitamin K content; pacemaker or platinum implantation). Participants with clinically elevated liver enzymes (more than three times the normal range's upper limit) were excluded since our primary endpoint included liver fat and steatohepatitis as a primary outcome.

We now include the relevant citation in the main text as suggested:

“Among 378 volunteers, 294 met age (30+ years of age) and abdominal obesity inclusion criteria [waist circumference (WC): men>102 cm, women>88 cm^33,34^] or dyslipidemia [TG>150 mg/dL and high-density-lipoprotein-cholesterol (HDL-c) ≤40 mg/dL for men, ≤50 mg/dL for women^32^]. Exclusion criteria were inability to perform physical activity; serum creatinine ≥2 mg/dL; serum alanine aminotransferase or aspartate aminotransferase more than three times above the upper limit of normal; a major illness that might require hospitalization; pregnancy or lactation; active cancer, or chemotherapy treatment in the last three years; warfarin treatment; pacemaker or platinum implantation; and participation in a different trial.”

2. The authors could consider including key details on the intervention study (those listed in SI Table 3), which are integral to the interpretations, in the main body of the text, rather than supplementary analyses (e.g., kcal restrictions). It would also be helpful if the authors could clarify how the randomisation to different intervention groups is accounted for in the current study.

We now include a summary of the key details of the intervention in Methods section 2.3:

“…all participants received free gym membership, including educational sessions encouraging moderate-intensity PA. Participants in both MED groups were assigned to a diet rich in vegetables, with poultry and fish replacing beef and lamb, with 1500-1800 kcal/day for men, 1200-1400 kcal/day for women. The diet additionally included 28 g/day of walnuts (+440 mg/day polyphenols provided). The green-MED group further consumed green tea (3–4 cups/day) and Wolffia globosa green shake (100 g/ day frozen cubes, +1240 mg/day total polyphenols provided).”

Regarding the randomization procedure, we refer the reviewer to Methods section 2.3:

“All participants completed the baseline measurements and were randomized, using a computer-based program, in a 1:1:1 ratio, stratified by sex and work status (to ensure equal workplace-related lifestyle features between groups), into one of the three intervention groups.”

Reviewer #3 (Recommendations for the authors):1) The authors did not choose to calculate brain age gap estimates for T0 and T18, but instead projected the brain-predicted age from T0 to T18 and calculated the difference between this 'expected brain age' and the 'observed brain age'. This procedure appears to be rather unusual. Is there any citeable resource that inspired the authors to follow this approach?

We applied the same approach here as in a previous study by our group where we tested methylation aging in the CENTRAL cohort (https://clinicaltrials.gov/ct2/show/NCT01530724):

Yaskolka Meir, A., Keller, M., Bernhart, S. H., Rinott, E., Tsaban, G., Zelicha, H., … & Shai, I. (2021). Lifestyle weight-loss intervention may attenuate methylation aging: the CENTRAL MRI randomized controlled trial. *Clinical epigenetics*, *13*(1), 1-10.

We now cite the relevant publication in sections Results 3.3 and Methods 2.10:

“The primary outcome of the current work was brain age attenuation quantified as the difference between the expected and observed brain age at T18 ^4^”

“Brain aging attenuation was quantified as the difference between the expected and observed brain age at T18 ^4^”

In response to reviewer #2, comment 2, we demonstrate that this method is identical to subtracting the predicted age at T18 from that of T0 following age bias correction while considering the exact time difference between the two scans.

2) Brain age was associated with several measures at baseline (Table 2), including chronological age. Usually, researchers in this field report on associations of the brain age gap, bias-corrected for chronological age. In other words, they report on the incremental associations of brain age with other variables, beyond the effects of chronological age. Without correcting for chronological age, associations of brain age with other variables are strongly driven by age. On this account, I feel that the results provided in Table 2 are not very informative with respect to the validity of the brain age estimates. The motivation for showing these statistics (message to the reader), as well as the choice of showing tertiles remains unclear to me.

We thank the reviewer for this comment, and we agree that the report of the tertiles was cumbersome. We now omitted it and we report the correlation of age, predicted age, and the bias-corrected brain age gap in Table 1 and SI Table 1 for the entire group.

Within the text we only report the correlation with the bias corrected brain age gap:

“In Table 1 we report the correlation between baseline characteristics and age, predicted age, and the brain-age gap (see SI Table 1 for additional measures). The brain age gap was computed as the difference between the predicted and observed age after regressing out the effect of age on the this gap (brain age bias correction ^11^). At baseline, brain-age gap was correlated with chemerin (r=0.22, p=0.029), and with obesity-associated measurements obtained by MRI including visceral abdominal tissue (VAT): r=0.23, p=0.02 and superficial subcutaneous fat (SSC): r=-0.25, p=0.014.”

3) I appreciate that the authors report a power analysis, but with the given information, it is not very comprehensible how the effect size of 0.79 was derived. I assume that this effect size estimate refers to Cohen's d!? Was it calculated based on the t-values and degrees of freedom from the '24-month post-surgery visits compared to baseline' reported in the cited resource? In general, effect sizes observed in single studies do not constitute very reliable estimates. In my view, a better option would be a sensitivity power analysis, which tests the probability (1-β) of different hypothetical effect sizes (e.g., d ranging from 0 to 1.5) to reach significance (α = 0.05) in a sample of respective size (N = 102; Lakens 2022, https://doi.org/10.1525/collabra.33267, see Figure 9).

We now include a derivation of our power calculation in SI section 11.3:

“2.11 Effect size in previous work

In a recent study^105^, the authors reported a significant decrease in δ age 12 months following bariatric surgery. We converted the reported t statistic (t=3.66, p< 0.001, df=85) to an effect size (Cohen's d = 0.79, r = 0.369) using an effect size calculator (https://lbecker.uccs.edu/). Using a sample size calculator (Python statsmodels.stats.power), we found that with an α of 0.05 and β > 0.05, a sample size > 90 is required. With the given sample size (n=102), the probability of failing to reject the null hypothesis under the alternative hypothesis (β, Type II error rate) is 3%. Alternatively, for a Type II error rate lower than 0.1 with the given sample size, an effect size of 0.664 (Cohen's d) is required.”

4) The study uses a variety of variables and it remains uncertain if all relevant variables have been mentioned. For instance, the different categories derived from the food frequency questionnaires have not been described sufficiently. How many categories have been tested for an association with changes in brain age? In order to increase transparency, I suggest providing the "21-word solution" by Simmons et al. (http://dx.doi.org/10.2139/ssrn.2160588): "We report how we determined our sample size, all data exclusions (if any), all manipulations, and all measures in the study." If necessary, this statement might be adjusted to ensure that it is accurate.

We now include the 21 words in the Methods section:

“In line with Simmons et al. (2012) 21-word solution ^36^, we report how we determined our sample size, all data exclusions, all manipulations, and all measures in the study.”

We now detail how all measures and FFQ variables were selected (Methods 2.6, 2.7):

“All clinical measures tested in the current study were selected a-priori from a large pool of variables taken in the DIRECT-PLUS trial^37^. These measures were taken from five pre-selected categories: 1. Anthropometry that includes BMI and WC; 2. Liver biomarkers that included AST, ALT, GGT, ALKP, FGF 21, and chemerin; 3. Glycemic markers, including glucose HOMA-IR and HbA1c; 4. Lipids including cholesterol, HDL-C, LDL-C, and triglycerides; 5. Imaging measures included liver fat, VAT, DSC, SSC, and the hippocampal occupancy score.”

“We selected a-priori the questionnaire variables that were associated with brain age attenuation. These variables included the change in the following categories: sweets and beverages, weekly *Wolffia globose* intake, nut and seeds, eggs and milk, beef, processed food, green tea, and walnuts.”

We now include a Table reporting all correlations of change in all reported food consumption variables to brain age attenuation in Supplementary file 4.